# Effect of *Sox18* on the Induction Ability of Dermal Papilla Cells in Hu Sheep

**DOI:** 10.3390/biology12010065

**Published:** 2022-12-30

**Authors:** Tingyan Hu, Xiaoyang Lv, Tesfaye Getachew, Joram M. Mwacharo, Aynalem Haile, Kai Quan, Yutao Li, Shanhe Wang, Wei Sun

**Affiliations:** 1College of Animal Science and Technology, Yangzhou University, Yangzhou 225009, China; 2International Joint Research Laboratory in Universities of Jiangsu Province of China for Domestic Animal Germplasm Resources and Genetic Improvement, Yangzhou University, Yangzhou 225009, China; 3Joint International Research Laboratory of Agriculture and Agri-Product Safety of Ministry of Education of China, Yangzhou University, Yangzhou 225009, China; 4International Centre for Agricultural Research in the Dry Areas, Addis Ababa 999047, Ethiopia; 5College of Animal Science and Technology, Henan University of Animal Husbandry and Economy, Zhengzhou 450060, China; 6CSIRO Agriculture and Food, 306 Carmody Rd, St Lucia 4067, Australia; 7“Innovative China” “Belt and Road” International Agricultural Technology Innovation Institute for Evaluation, Protection, and Improvement on Sheep Genetic Resource, Yangzhou University, Yangzhou 225009, China

**Keywords:** *Sox18*, Hu sheep, dermal papilla cells, induction capacity

## Abstract

**Simple Summary:**

The *Sox18* gene is important for the growth and function of dermal papilla cells, but its effect on the induction ability of dermal papilla cells in Hu sheep is not clear. Through experiments, we analyzed the effect of *Sox18* on the induction ability of dermal papilla cells in Hu sheep. The results showed that *Sox18* could enhance the induction ability of dermal papilla cells, and regulate the induction ability of dermal papilla cells through the Wnt/β-catenin signal pathway. We believe that *Sox18* is an important gene that affects the induction ability of Hu sheep’s dermal papilla cells, and these findings may provide new avenues for studying the growth and development mechanism of hair.

**Abstract:**

*Sox18* is a developmental gene that encodes transcription factors. It has been indicated as be a key gene affecting the growth and development of hair follicles, in which dermal papilla cells (DPCs) have been demonstrated to play an important role through their ability to induce the formation of hair follicles. Pre-laboratory studies have found that *Sox18* is differentially expressed in the dermal papilla cells of different pattern types of Hu sheep. We speculated that *Sox18* plays an important role in the dermal papilla cells of Hu sheep. In our study, we analyzed the effect of *Sox18* on the induction ability of DPCs in order to elucidate the function and molecular mechanism of *Sox18* in the DPCs of Hu sheep. We first identified the expression of *Sox18* in the DPCs of Hu sheep by immunofluorescence staining. We then used alkaline phosphatase staining, cell morphology observations and RT-PCR to detect the effect of *Sox18* on the induction of DPCs after overexpression of or interference with *Sox18*. We also used RT-PCR, WB and immunofluorescence staining to detect the effect of *Sox18* on the Wnt/β-catenin signal pathway in DPCs. We found that *Sox18* was specifically expressed in the DPCs of Hu sheep, and that *Sox18* could enhance the alkaline phosphatase activity in the DPCs of Hu sheep and accelerate cell agglutination. The results of RT-PCR revealed that *Sox18* promoted the mRNA expression of *Versican*, *HHIP* and *FGFRI*, and inhibited the mRNA expression of *BMP4* and *WIF1*. Further studies showed that *Sox18* promoted the expression of *β-catenin* and activated the Wnt/β-catenin signal pathway in DPCs. When the Wnt/β-catenin signal pathway of DPCs was activated, the induction ability of DPCs was enhanced. Overall, we believe that *Sox18* could enhance the induction ability of DPCs in Hu sheep and regulate the induction ability of DPCs through the Wnt/β-catenin signal pathway.

## 1. Introduction

The Hu sheep is a rare breed in China and is best known for its white lambskins with a wavy pattern. The lambskins have four grades: small waves, medium waves, large waves and straight wool. Among these, the pattern of small waves is the best and straight wool is the worst [1]. The curvature of wool is closely related to the development of the hair follicles, and dermal papilla cells (DPCs) can express and secrete a variety of growth factors and signaling molecules to stimulate the proliferation and differentiation of hair-producing cells, thus regulating the growth and development of hair follicles [2]. Driskell et al. found that the number of DPCs and their gene expression patterns can influence hair type and hair curvature [3], and Takahashi et al. showed that the Wnt/β-catenin signaling pathway has an influential function within the process of hair growth by regulating the induction ability of DPCs [4].

The hair follicles are important accessory organs of the skin that have a complex morphology that controls the growth of the mammalian coat and has a unique regenerative function [5]. After the birth of mammals, the growth of hair follicles has the characteristics of a lifetime cycle: a growth cycle consisting of the anagen, catagen and telogen phase. DPCs are a group of specialized fibroblasts that grow in an agglomeration at the basal layer of the hair follicle and are surrounded by hair-producing cells, and their most important characteristic is that they can maintain and induce hair follicles’ growth and development. A study has shown that hair follicles stop growing immediately after removing DPCs in the anagen phase, and transplanting DPCs in anagen or low-generation DPC lines with hair-producing cells into mouse ears can induce the formation of new hair follicles [6]. When DPCs were cultured in vitro, it was found that DPCs still had the ability to induce the formation of hair follicle [7,8], and had the characteristic of agglutinative growth which could form multi-layer agglutinates [9]. This growth characteristic seems to be closely related to the ability of DPCs to induce the formation of hair follicles [10,11]. DPCs have the function of inducing the formation of hair follicles and regulate their growth and development. Therefore, the induction ability of DPCs has a major effect on hair follicles and hair. Some growth factors secreted by DPCs participate in the regulation of periodic changes in the hair follicles, mediated the signal interaction between the dermis and epidermis of hair follicles, and promote the proliferation and differentiation of hair follicle cells, thus controlling the growth and development of hair follicles [12]. On the one hand, some characteristic molecules expressed by DPCs have been considered as typical indicators of periodic hair follicle growth, such as alkaline phosphatase (*ALP*), *Versican*, etc. [13,14,15,16]. In addition, some genes play an important role in maintaining the induction characteristics of DPCs, such as insulin-like growth factor 1 (*IGF1*), fibroblast growth factor 1 (*FGF1*), bone morphogenetic protein 4 (*BMP4*) and *Wnt3a* [17,18,19,20,21,22].

*Sox18* is located on chromosome 13 in sheep. In the early stage of our laboratory work, through 10 × single-cell transcriptome genomic sequencing, we identified different types of cells in Hu sheep’s hair follicles and revealed the differential genes in different pattern types of hair follicles [23]. It was found that *Sox18* was specifically expressed in the DPCs of Hu sheep and were significantly differentially expressed in the DPCs of different pattern types in Hu sheep [23]. Previous studies have found that nonsense mutations of *Sox18* could cause hair development defects in mice. The heterozygotes shown a reduction in curved “zigzag hairs”, and the homozygotes showed a lack of whiskers and coat [24]. In *Sox18* mutant mice, the expression of *Wnt5a* and the activity of the Wnt signaling pathway decreased significantly, indicating that *Sox18* was a key gene that regulated the Wnt signaling pathway and affected the hair type and hair curvature in mice [25]. Combined with the function of *Sox18* in the development of hair follicles in mice and its regulation of the Wnt signaling pathway, it was speculated that *Sox18* affects the function of DPCs by regulating Wnt signaling in DPCs, and then plays an important role in the regulating the formation of lambskin patterns, but its specific regulatory mechanism still needs to be explored in depth. In this study, we will analyze the effect of *Sox18* on the induction ability of DPCs in Hu sheep by overexpressing and interfering with *Sox18* in the DPCs of Hu sheep. Ultimately, our results will provide a new direction for studying the molecular mechanism of lambskin pattern formation.

## 2. Materials and Methods

### 2.1. Culture of DPCs of Hu Sheep

All DPCs of Hu sheep used for the experiments were previously isolated and cultured in our laboratory [26]. The DPCs were cultured in DMEM/F12 (HyClone, Logan, UT, USA) supplemented with 10% fetal bovine serum (Gibco, Grand Island, NY, USA) and 1% penicillin–streptomycin (Gibco, Grand Island, NY, USA) at 37 °C with 5% CO_2_.

We used a 0.25% trypsin-EDTA solution (Solarbio, Beijing, China) to digest the DPCs and considered each digestion of DPCs as one cell passing through to the next generation. The third-generation DPCs had a single morphology and good growth activity, so the third-generation DPCs were used in the experiment.

### 2.2. Immunofluorescence Staining in Dermal Papilla Cells of Hu Sheep

The cells were fixed in 4% formaldehyde and incubated in primary antibodies (1:200) and secondary antibodies (1:200). The DPCs were stained by adding Hoechst dye (1 μg/mL), and the images were taken by a fluorescence inversion microscope system (Nikon, Tokyo, Japan). Primary antibodies included those against VIM (sc-6260, Santa Cruz Biotechnology, Dallas, TX, USA), β-catenin (66379-1-Ig, Proteintech, Wuhan, China), Versican (D223532, Sangon, Shanghai, China), and Sox18 (DF8720, Affinity Biosciences, Changzhou, China). Secondary antibodies included goat anti-rabbit IgG H&L (ab150078, Abcam, Cambridge, UK) and goat anti-mouse IgG H&L (ab150114, Abcam, Cambridge, UK).

### 2.3. Total Cellular RNA Extraction

When the DPCs were up to 80–90% confluence in six-well plates, or 24 h after cell transfection was completed, Trizol Universal Reagent (Tiangen, Beijing, China) was used to extracted total cellular RNA according to its instruction, and the RNA was stored at −80 °C until use.

### 2.4. Construction of the Sox18 Overexpression Vector

#### 2.4.1. Full-Length Amplification of the Sox18 CDS Region and Restriction of pcDNA3.1 Plasmid

According to NCBI (https://www.ncbi.nlm.nih.gov/ (accessed on 17 January 2022)), the full-length CDS (coding sequence) of the *Sox18* gene of sheep (1176 bp, GenBank Accession: XM_027976914.2) was acquired. The amplification primers were then designed by using Primer Premier 5. The primer sequences were as follows:

F: ctagcgtttaaacttaagcttATGCAGAGATCGCCGCTCGG

R: ccacactggactagtggatccCTATCCAGAGATGCAGGCGCTGTAG

The uppercase letters indicate the full-length amplified CDS of *Sox18*. The lowercase letters indicate the terminal sequence of the linearized vector (pcDNA3.1(+)) for homologous recombination, and the lowercase letters that have been underlined indicate the restriction enzyme digestion sites of HindIII (aagctt) and BamHI (ggatcc).

The full-length *Sox18* CDS was amplified by using Prime STAR Max DNA polymerase reagent (Takara, Beijing, China), and the PCR products were identified by 1% agarose gel electrophoresis. The PCR products were recovered by using a SanPrep Column PCR Product Purification Kit (Sangon, Shanghai, China), and the DNA concentration was calculated by a micro-ultraviolet spectrophotometer (Life Real, Hangzhou, China).

The restriction enzymes HindⅢ and BamHIII (Takara, Kusatsu, Shiga, Japan) were used to cut the pcDNA3.1(+) plasmid (Youbio, Changsha, China), and the enzyme’s digestion products were detected with 1% agarose gel electrophoresis. The enzyme digestion products were then recovered and measured. Finally, all recovered products were stored at −20 °C.

#### 2.4.2. Ligation of the Sox18 Fragment with Linear Vector

According to the manual of ClonExpress^®^ II One Step Cloning Kit, the recombination reaction was carried out. The recombinant product was then transformed into Trelief™ 5α chemically competent cells (Tsingke, Beijing, China). Individual colonies were selected and placed in tubes with 15 mL of ampicillin-resistant LB broth and shaken for 16 h. The plasmid was extracted from 10 mL of the bacterial solution by using an EndoFree Mini Plasmid Kit Ⅱ (Tsingke, Beijing, China) and was examined by double restriction endonuclease digestion. Finally, the successfully constructed vector was named pcDNA3.1(+)-Sox18 (Appendix A) and stored at −20 °C.

### 2.5. Sox18 siRNA Synthesis

The siRNA of *Sox18* was purchased from Shanghai GenePharma Co., Ltd. (Shanghai, China). The sequences are shown in Table 1.

### 2.6. Cell Transfection

When the DPCs had grown to 50–70% confluence, the plasmids or siRNAs were transfected into the DPCs by using jetPRIME (Polyplus transfection, Illkirch, France). 

### 2.7. Alkaline Phosphatase Activity

The DPCs were fixed with 4% formaldehyde. The alkaline phosphatase activity of DPCs was detected by using a BCIP/NBT Alkaline Phosphatase Color Development Kit (Beyotime, Shanghai, China). Finally, the images were acquired by a fluorescence inversion microscope system (Nikon, Tokyo, Japan) and we used Image J software to calculate the staining area in different treatment groups from the same field of view.

### 2.8. Real-Time Quantitative Polymerase Chain Reaction (RT-PCR)

FastKing gDNA Dispelling RT SuperMix (TIANGEN, Beijing, China) was used to obtain the first-stand cDNA, then the mRNA expression levels were detected by 2×TSINGKE^®^ Master qPCR Mix (SYBR Green I) (Tsingke, Beijing, China).

The primers’ information is listed in Table 2. We used GAPDH as the reference genes and each RT-PCR experiment included three biological replicates and three technical replicates.

### 2.9. Western Blotting

The protein was extracted using a Total Protein Extraction Kit (Beyotime, Shanghai, China), and the concentration was measured using a BCA Protein Assay Kit (Beyotime, Shanghai, China). The proteins were separated and transferred to PVDF membranes. The PVDF membranes were probed with the primary antibodies and the secondary antibodies. Finally, the expression of protein was measured using the ECL Western Blot kit (Biosharp, Hefei, China) and analyzed by a ChemiDocTM Touch Imaging System (Bio-Rad, CAL, USA).

The primary antibodies were anti-Sox18 (1:1000, DF8720, Affinity Biosciences, Changzhou, China), anti-β-catenin (1:1000, AF0069, Beyotime, Shanghai, China) and anti-GAPDH (1:3000, T0004, Affinity Bio-sciences, Changzhou, China). The secondary antibodies were HRP goat anti-rabbit IgG (1:1000, AS014, ABclonal, Wuhan, China) and HRP goat anti-mouse IgG (1:1000, AS014, ABclonal, Wuhan, China).

### 2.10. Activation and Inhibition of Wnt/β-Catenin Signal Pathway

SKL2001 (HY-101085, MedChemExpress, Shanghai, China) was used as an agonist of the Wnt/β-catenin signal pathway and IWP-2 (SF6831, Beyotime, Shanghai, China) was used as an inhibitor of the Wnt/β-catenin signal pathway [27,28]. The DPCs were grown to 50–80% confluence and treated with 40 μΜ SKL2001 or 10 μΜ IWP-2 for 24 h. 

### 2.11. Statistical Analysis

SPSS 20.0 software was used for statistical analysis. Independent samples *t*-tests were used to calculate the statistical significance between two groups, and one-way ANOVA was used for multiple groups. Composite data are shown as the mean ± standard error.

## 3. Results

### 3.1. Sox18 Was a Marker Gene of DPCs in Hu Sheep

After the isolation and culture of DPCs, immunofluorescence identification was carried out. Firstly, the expression of the specific marker genes *VIM* and *Versican* in the DPCs of Hu sheep was detected, and the results were positive (Figure 1). It was indicated that the cells cultured in this experiment were dermal papilla cells. In combination with the result of 10× single cell transcriptome genomic sequencing, which showed that *Sox18* was specifically expressed in the DPCs of Hu sheep, the expression of *Sox18* in the primary DPCs of Hu sheep was detected. The result showed that all cells were positive (Figure 1), which indicated that *Sox18* was a marker gene of DPCs in Hu sheep.

### 3.2. Sox18 Enhanced the Induction Ability of DPCs in Hu Seep

In order to express *Sox18* efficiently in the DPCs of Hu sheep and explored the effect of *Sox18* on the induction ability of DPCs in Hu sheep, we constructed a *Sox18* overexpression vector successfully and transfected it into the DPCs (Figure 2).

Alkaline phosphatase activity is an important index for evaluating the induction ability of DPCs [15]. Therefore, we detected the alkaline phosphatase activity of DPCs in Hu sheep by staining, and an increased number of staining sites and a darker color indicated stronger alkaline phosphatase activity. Firstly, we detected the alkaline phosphatase activity of primary DPCs. The results showed that primary DPCs had strong alkaline phosphatase activity (Figure 3A). We then stained the third-generation DPCs and found that the alkaline phosphatase activity of the third-generation DPCs became weak (Figure 3A). Finally, we transfected pcDNA3.1(+)-*Sox18*, pcDNA3.1(+), siRNA-*Sox18* and siRNA-NC separately in the third-generation DPCs, and detected alkaline phosphatase activity in the DPCs 24 h after transfection. The results showed that overexpression of *Sox18* significantly enhanced the alkaline phosphatase activity of DPCs, and interfering with *Sox18* decreased the alkaline phosphatase activity of DPCs (Figure 3B,C). Further observations showed that the overexpression of *Sox18* could promote the agglutination of DPCs (Figure 3D). These results showed that *Sox18* enhanced the induction ability of DPCs.

The mRNA expression levels of the target gene *Sox18* and the genes related to the induction ability of DPCs were detected by RT-PCR. *GAPDH* was the internal reference gene. After overexpression of *Sox18* in DPCs, the mRNA expression of *Sox18* increased significantly (*p* < 0.05) (Figure 4A). Moreover, the mRNA expression levels of *Versican*, *HHIP* and *FGFRI* increased, and the mRNA expression levels of *BMP4* and *WIF1* decreased (Figure 4B). However, after interfering with *Sox18* in DPCs, the mRNA expression of *Sox18* decreased significantly (*p* < 0.05) (Figure 4C). At the same time, the mRNA expression levels of *Versican*, *HHIP* and *FGFRI* decreased, and the mRNA expression levels of *BMP4* and *WIF1* increased (Figure 4D). The results of Western blotting showed that the protein expression levels of *Sox18* were upregulated after overexpression of *Sox18* (Figure 4E), and decreased after interfering with *Sox18* (Figure 4F). To sum up, Sox18 could enhance the induction ability of DPCs.

### 3.3. Sox18 Regulated the Induction Ability of DPCs through the Wnt/β-Catenin Signal Pathway

The Wnt/β-catenin signal pathway is closely related to the growth and development of hair follicles. The results of RT-PCR and WB revealed that the expression of *β-catenin* was significantly changed in DPCs when *Sox18* was overexpressed or interfered with (Figure 5A,B). We speculated that *Sox18* regulated the induction ability of DPCs through the Wnt/β-catenin signal pathway. SKL2001 was used to activate the Wnt/β-catenin signal pathway of DPCs [27]. After the Wnt/β-catenin pathway signal of DPCs had been activated, the amount of β-catenin expressed in the nucleus of DPCs increased (Figure 5C). When *Sox18* was overexpressed in DPCs, the amount of β-catenin expressed in the nucleus of DPCs also increased, indicating that *Sox18* activated the Wnt/β-catenin signal pathway of DPCs (Figure 5C). Further study found that the alkaline phosphatase activity of DPCs was enhanced when *Sox18* was overexpressed in DPCs, which was the same as when the Wnt/β-catenin signal pathway was activated (Figure 5D,E). However, the positive effect of *Sox18* on the alkaline phosphatase activity of DPCs was aborted when we added IWP-2, an inhibitor of the Wnt/β-catenin signal pathway [28], to DPCs overexpressing *Sox18* (Figure 5D,E). Therefore, we believe that the Wnt/β-catenin signal pathway was closely related to the induction ability of DPCs. It was further observed that overexpressed *Sox18* as well as activation of the Wnt/β-catenin signal pathway in DPCs enhanced the agglutination of DPCs, while inhibition of the Wnt/β-catenin signal pathway weakened the effect of *Sox18* on the agglutination of DPCs (Figure 5F). This further suggested that *Sox18* regulated the induction ability of DPCs through the Wnt/β-catenin signal pathway. To sum up, *Sox18* activated the Wnt/β-catenin signal pathway in DPCs, thus enhancing the induction ability of DPCs.

## 4. Discussion

DPCs have the ability to induce the formation of hair follicles and regulate their growth. When cultured in vitro, DPCs still had the ability to induce hair follicle formation [7,8], and had the characteristic of agglutinative growth that allowed the formation of multi-layer agglutinates [9]. This growth characteristic seems to be closely related to the ability of DPCs to induce hair follicle formation [10,11]. However, as the number of generations increased, the ability and growth characteristics are gradually reduced and were ultimately lost, resulting in inactivation of the hair follicle and causing hair loss [29,30,31].

Early studies have shown that *Sox* (SRY-related high mobility group box) genes are involved in a variety of early embryonic developmental processes such as blood cell production, sex determination, skeletal development, and nervous system and lens development [32,33]. *Sox18* is a key gene for the growth and development of hair follicles in mice. The expression of *Sox18* has been detected in DPCs, and it plays an important role in the development and function of DPCs [24,34,35]. Villani et al. constructed mouse models of *Sox18* mutations, including *Sox18*^−/−^, *Sox18*^+/Ra^ and *Sox18*^Op/+^, in which *Sox18*^Op/+^ mutants exhibited severe hair development defects [25]. The dominant negative mutation of *Sox18* inhibited the formation and differentiation of DPCs, resulting in defective hair growth and development. *Sox18* mutation also prevented newborn dermal cells or hair papillae form inducing hair formation in regeneration experiments. Further analysis using gene chips indicated that *Sox18* may affect the Wnt signaling pathway in epidermal cells by regulating the expression of *Wnt5a*, which, in turn, affects the hair’s morphology [25]. *Sox18* has the function of regulating the normal differentiation of DPCs in all hair types, and *Sox18* acts as a mesenchymal molecular switch that is required for the formation and function of DPCs in all hair types [36]. *Sox18* plays an important role in the growth and development of hair follicles, but the molecular mechanism by which the Sox18 affects wool curvature in sheep needs to be further studied.

Some growth factors secreted by DPCs can participate in the regulation of periodic changes in the hair follicles through autocrine or paracrine secretion. These growth factors mediate the signaling interaction between the dermal and epidermal parts of hair follicles, and they promote the proliferation and differentiation of hair follicle cells, thus controlling the growth and development of hair follicles [12]. ALP, which is secreted by DPCs, is a hydrolase that releases phosphate groups in a mild alkaline environment. The formation of hair follicles can be more effectively induced when DPCs with higher ALP activity are transplanted into a skin wound of a recipient [37]. Moreover, Iida M et al. also confirmed that the ALP activity of dermal papilla cells changed with the periodic growth of hair follicles, with the highest activity in the anagen phase and the lowest in the telogen phase [15]. The level of ALP activity in DPCs is related to the formation of hair follicles, so ALP is considered to be an index for evaluating the periodic growth activity of hair follicles induced by DPCs. The expression of *Versican* was detected in the dorsal hair follicles of mice from the early to mid-anagen phase, but it was found in only in a few hair follicles’ DPCs during the late anagen phaes, and none was found in hair papillae during the catagen and telogen phases [38]. This suggests that *Versican* is a key regulator in the anagen phase. In addition, *Versican* plays an important role in maintaining the processes of cell proliferation, differentiation and migration during the anagen phase [39]. *BMP4* is a member of the BMPs family and acts mainly at the end of the anagen phase and in the process of entering the telogen phase [19]. *BMP4* can inhibit the transformation of hair follicles from the telogen to the anagen phase, and regulates the shape, size and position of hair follicles [40]. Wnt inhibitory factor 1 (*WIF1*) is the suppressor gene of the Wnt/β-catenin signaling pathway, which blocks the downstream the Wnt signal pathway by directly binding to the Wnt ligand and preventing it from binding to the complex of the coiled-coil protein receptor and LRP5/6 [41]. *WIF1* is considered to be the marker gene of dermal condensate, and its expression is significantly upregulated during the induction of primary hair follicles in Tibetan sheep, which may regulate the formation of primary hair follicles’ condensate in Tibetan sheep [42]. *HHIP* is considered to be a potential antagonist of *SHH*, which inhibits the activation of the SHH signaling pathway through a negative feedback mechanism, and SHH signaling transduction is very important for hair follicle development [43]. *FGFR1* is a transmembrane receptor of some fibroblast growth factors that has tyrosine kinase activity and is located in the papilla of hair follicles to promote the growth and development of hair follicles [44]. The results of this study are consistent with the reported results.

The normal development and regular growth of hair follicles depend on the regulation of a complex network of signaling pathways between the dermis and keratinocytes, including the Wnt signal pathway, the BMP signal pathway, the SHH signal pathway and the Notch signal pathway [45,46,47,48,49,50]. The Wnt signal pathway is very important for controlling the number, size, growth and regeneration of hair follicles [51]. The Wnt signal pathway includes four branches, and the Wnt/β-catenin signal pathway is the one most closely related to the growth of hair follicles [52]. After human dermal papilla cells were treated with *Wnt3a*, the expression of β-catenin signaling was increased and β-catenin signaling was attenuated after treatment with *Wnt5a*. Moreover, transfection with *Wnt5a*-siRNA in cultured DPCs enhances β-catenin signaling [53]. In addition to Wnt protein, *β-catenin* can also regulate the hair growth cycle through the Wnt/β-catenin signal pathway. *β-catenin* protein can regulate the formation of hair follicle substrate [54], and *β-catenin*, *FGF* and *IGF* in the dermal papillae can mediate the induction of the dermal papillae [55]. The key function of the Wnt signal pathway in hair follicle formation and hair growth has been widely recognized, but the specific mechanism of the Wnt/β-catenin signal pathway involved in the growth and development of hair follicles needs to be further studied. In our study, we found that *Sox18* could activate the Wnt/β-catenin signal pathway in DPCs and enhance the induction ability of DPCs. We speculated that *Sox18* regulated the induction ability of DPCs through the Wnt/β-catenin signal pathway, but the specific molecular mechanism still needs further study.

## 5. Conclusions

To sum up, this study verified that *Sox18* could improve the induction ability of DPCs in Hu sheep and did so through the Wnt/β-catenin signal pathway. These findings might help us to study the molecular mechanism of lambskin pattern formation in more depth.

## Figures and Tables

**Figure 1 biology-12-00065-f001:**
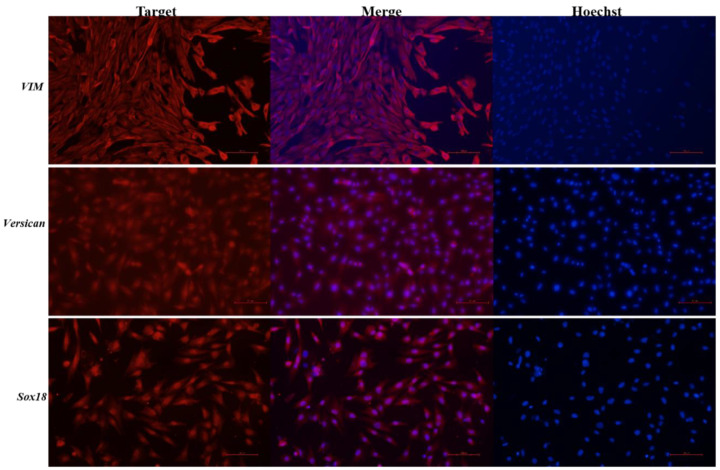
Immunofluorescence analysis of the expression of *VIM*, *Versican* and *Sox18* in the isolated DPCs. Scale bars, 100 μm. Red fluorescence indicates the nuclear expression pattern of the target protein. The nucleus was stained with blue Hoechst dye.

**Figure 2 biology-12-00065-f002:**
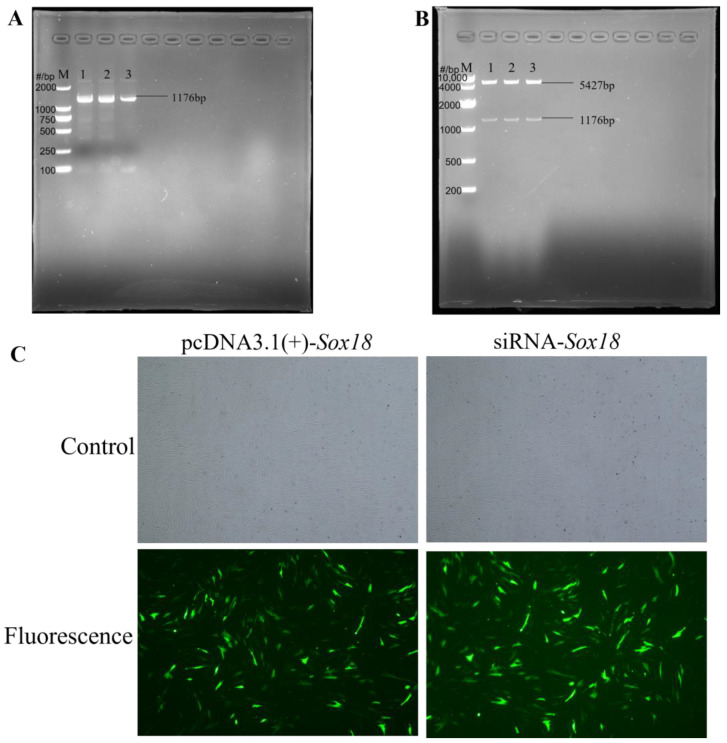
Construction of pcDNA3.1(+)-*Sox18*. (**A**) Full-length amplification of *Sox18* CDS from Hu sheep. Lane M is the DL2,000 marker; Lanes 1–3 are the PCR products. (**B**) Double restriction endonuclease digestion of pcDNA3.1(+)-*Sox18*. Lane M is the DL10,000 marker; Lanes 1–3 are the PCR products. (**C**) The transfection efficiency of pcDNA3.1(+)-*Sox18* and siRNA-*Sox18* in DPCs from Hu sheep. Scale bars, 500 μm.

**Figure 3 biology-12-00065-f003:**
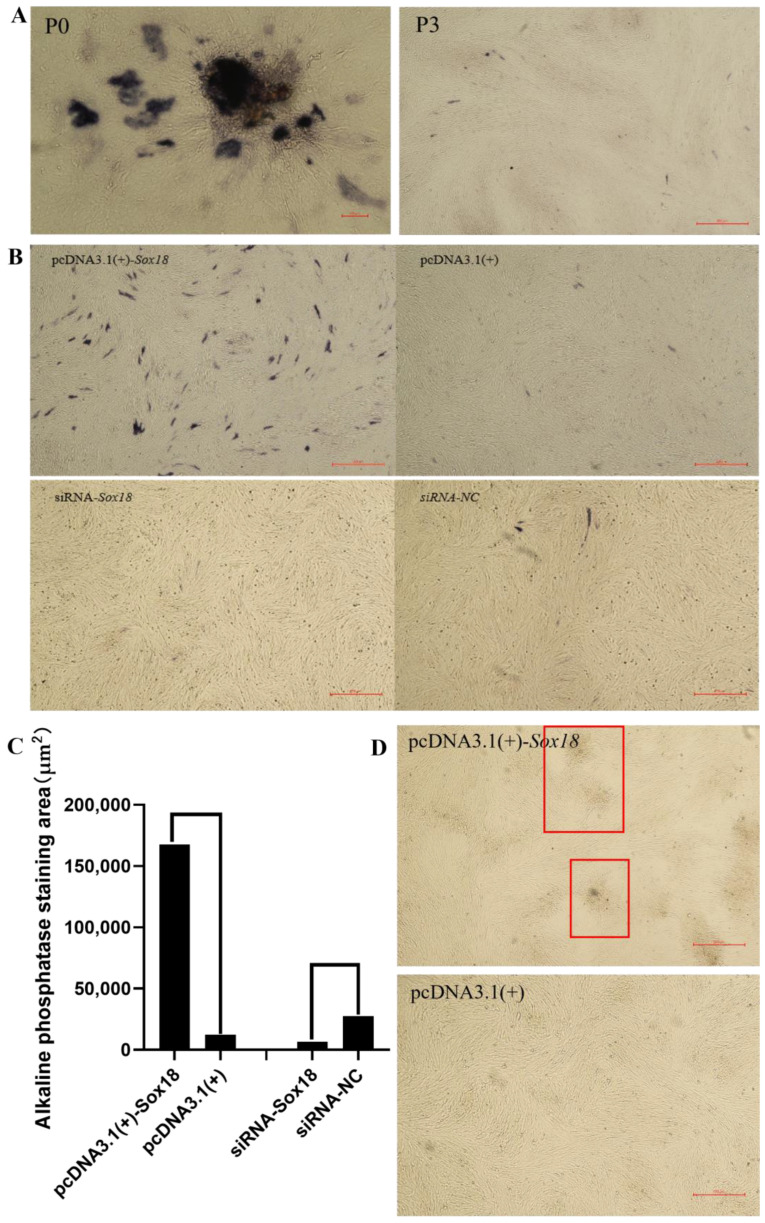
The alkaline phosphatase activity of DPCs. (**A**) The primary DPCs and the third-generation DPCs. (**B**) The third-generation DPCs transfected with pcDNA3.1(+)-*Sox18*, pcDNA3.1(+), siRNA-*Sox18* and siRNA-NC for 24 h. (**C**) The alkaline phosphatase staining area. (**D**) The agglutination of DPCs. Scale bars, 500 μm.

**Figure 4 biology-12-00065-f004:**
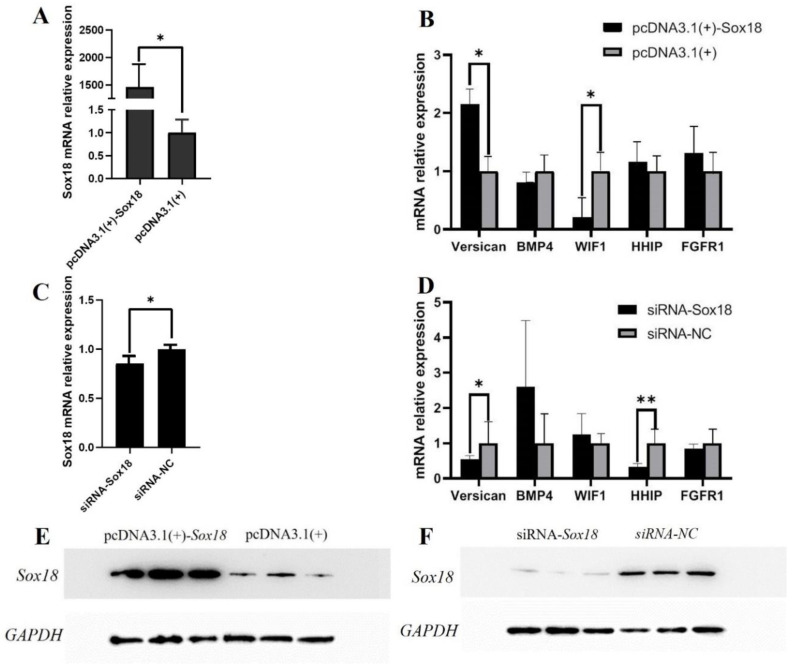
Effect of *Sox18* on the induction ability of DPCs. (**A**–**D**) The relative mRNA expression of *Sox18*, *Versican*, *BMP4*, *WIF1*, *HHIP* and *FGFRI*. (**E**,**F**) Detection of *Sox18* protein expression levels. “*” indicates a significant difference (*p* < 0.05) and “**” indicates a highly significant difference (*p* < 0.01).

**Figure 5 biology-12-00065-f005:**
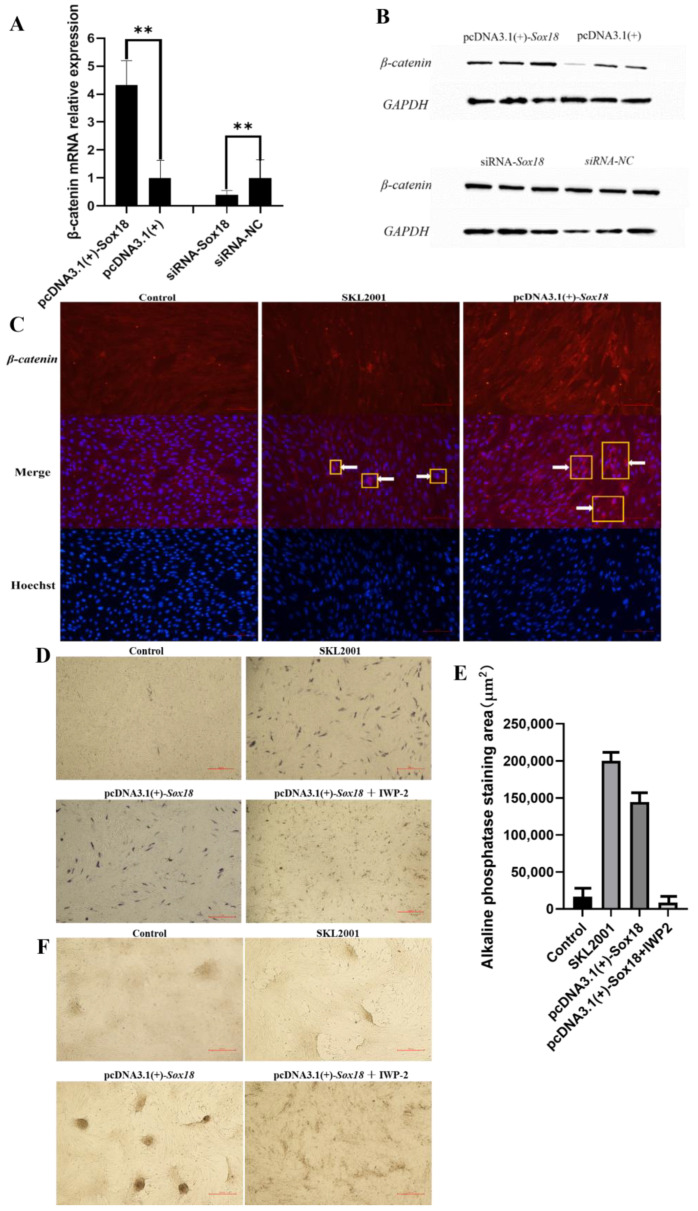
Effect of *Sox18* on the Wnt/β-catenin signal pathway in DPCs. (**A**)The relative mRNA expression of *β-catenin*. (**B**) Detection of the expression levels of *β-catenin* protein. (**C**) Immunofluorescence analysis of the expression of *β-catenin* in DPCs. (**D**) The alkaline phosphatase activity of DPCs. (**E**) The alkaline phosphatase staining area. (**F**) The agglutination of DPCs. “**” indicates a highly significant difference (*p* < 0.01). Scale bars, 100 μm or 500 μm.

**Table 1 biology-12-00065-t001:** Sequences of *Sox18* siRNA and NC.

Name	Sequences (5′→3′)
siRNA-*Sox18*	Sense:ACCAGUACCUCAACUGCAGTT
	Antisense:CUGCAGUUGAGGUACUGGUTT
siRNA-NC	Sense:UUCUCCGAACGUGUCACGUTT
	Antisense:ACGUGACACGUUCGGAGAATT

**Table 2 biology-12-00065-t002:** The information of the primers used for RT-PCR.

Name	Forward Primer (5′-3′)	Reverse Primer (5′-3′)	Product Length (bp)
*Sox18*	TGTGGGCGAAGGACGAGC	GCCAAGCCTGGGAGGAGGAG	253
*Versican*	TACAAAGGGAGGGTGTCGGT	AAGCCTTCTGTGCCATCTCA	226
*β-catenin*	TGAGTGGGAACAGGGGTTTT	TGAGCAGCATCGAACTGTGT	162
*BMP4*	TAGCAAGAGCGCAGTCATCC	AGCAGAGTTTTCGCTGGTCC	196
*WIF1*	AAGCCTGTACCTGTGGATCG	CTGGCATTCTCTGCTGTGCT	134
*HHIP*	GTGGCCTGTGCTTTCCTGAT	AGAATGAAGAGGCGGTGGGA	208
*FGFR1*	CCCGGAGATGTTGCCATCAA	GCCCTGTTCCTCTTTGCCAT	135
*GAPDH*	TCTCAAGGGCATTCTAGGCTAC	GCCGAATTCATTGTCGTACCAG	151

## Data Availability

The raw data supporting the conclusions of this article will be made available by the authors, without undue reservation. The single-cell RNA sequencing data mentioned in this research have been deposited in the NCBI’s Gene Expression Omnibus database (https://www.ncbi.nlm.nih.gov/geo/ (accessed on 20 October 2021)) under accession number: GSE186204.

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
