# Peer review of "Effect of *Sox18* on the Induction Ability of Dermal Papilla Cells in Hu Sheep"

_biology, 2022, doi:10.3390/biology12010065_

Round 1

Reviewer 1 Report

Manuscript ID: biology-2059078

Title: Effect of Sox18 on the Induction Ability of Dermal Papilla Cells in Hu Sheep

The work is a good attempt at trying to evidence the effect of Sonx18 on the induction ability of Dermal Papilla Cells (DPCs) in Hu Sheep. For this, the authors used immunofluorescence and alkaline phosphatase staining, cell morphology observation, and RT-PCR to detect the effect of Sox18 on the induction of DPCs after overexpression or interference with Sox18.  Due to the evidence, it was concluded that Sox18 could enhance the induction ability of DPCs in Hu sheep and regulated the induction ability of DPCs through Wnt/β-catenin signal pathway. Although, however, it was not possible demonstrate the molecular mechanism by which the Sox18 affects wool curvature in sheep, as mentioned in the introduction. Thus, as mentioned by the authors, there is a need to be further studies.

The work is ACCEPTABLE. There are no comments against the publication of the article. However, some specific comments are addressed to the authors.

Specific Comments

1 – In the introduction (somewhere in between lines 81 and 85, for example) it would be useful to mention on which chromosome the Sox18 gene is located.

2- The same sentence is repeated between lines 104 and 106.

3 – Closing parentheses in line 121.

4 – Maybe is not necessary, but it would be useful to put the meaning of CDS in line 125 (coding sequence).

5 – In lines 129 and 130 show the lowercase underline for both Forward and Reverse primer.

Author Response

Response to Reviewer 1 Comments:

Point 1: The work is a good attempt at trying to evidence the effect of Sonx18 on the induction ability of Dermal Papilla Cells (DPCs) in Hu Sheep. For this, the authors used immunofluorescence and alkaline phosphatase staining, cell morphology observation, and RT-PCR to detect the effect of Sox18 on the induction of DPCs after overexpression or interference with Sox18.  Due to the evidence, it was concluded that Sox18 could enhance the induction ability of DPCs in Hu sheep and regulated the induction ability of DPCs through Wnt/β-catenin signal pathway. Although, however, it was not possible demonstrate the molecular mechanism by which the Sox18 affects wool curvature in sheep, as mentioned in the introduction. Thus, as mentioned by the authors, there is a need to be further studies.

The work is ACCEPTABLE. There are no comments against the publication of the article. However, some specific comments are addressed to the authors.

Response 1: Thanks for your comments and we thank for your work. We will further verify the effect of Sox18 on the induction ability of Dermal Papilla Cells in Hu Sheep and the function of Sox18 in the molecular mechanism of wool curvature.

Point 2: Specific Comments

1 – In the introduction (somewhere in between lines 81 and 85, for example) it would be useful to mention on which chromosome the Sox18 gene is located.

2- The same sentence is repeated between lines 104 and 106.

3 – Closing parentheses in line 121.

4 – Maybe is not necessary, but it would be useful to put the meaning of CDS in line 125 (coding sequence).

5 – In lines 129 and 130 show the lowercase underline for both Forward and Reverse primer.

Response 2: Thanks for your comments, and we have revised as requested.

Reviewer 2 Report

The authors showed the regulatory mechanism and induction of  DPC in Hu sheep by Sox18. The experimental design and results were simple, but the data shows clearly. I find the data convincing. I do not have any substantial amendments to suggest.

This might be an exciting topic for publication in this journal.

Major concern;

In the DPCs induction experiments, the authors should be studied how many days of Sox18 expression levels are required by using tet-on inducible gene expression system. Also, the extent to which the induced DPCs retain ALP activity should be studied.

Minor concern;

1.The authors should show the transfection efficiency in the experimental conditions.

2.The number of ALP colonies should be quantified and compared to the control.

L116: no information for secondary antibody for immunostain, callar conjugation.

L148:Figure1 must change to constructed vector pcDNA3.1(+)-Sox18.

Figure6-C. Target should change to β-catenin.

Author Response

Point 1: The authors showed the regulatory mechanism and induction of DPC in Hu sheep by Sox18. The experimental design and results were simple, but the data shows clearly. I find the data convincing. I do not have any substantial amendments to suggest.

This might be an exciting topic for publication in this journal.

Response 1: Thanks for your comments and we thank for your work.

Point 2: Major concern;

In the DPCs induction experiments, the authors should be studied how many days of Sox18 expression levels are required by using tet-on inducible gene expression system. Also, the extent to which the induced DPCs retain ALP activity should be studied.

Response 2: Thanks for your comments. However, we are very sorry that our lab does not currently have a tet-on inducible gene expression system. We will attempt to use the tet-on inducible gene expression system to induce the expression of Sox18, and determine how many days of Sox18 expression levels are required and to what extent DPCs retain ALP activity.

Point 3: Minor concern;

1.The authors should show the transfection efficiency in the experimental conditions.

2.The number of ALP colonies should be quantified and compared to the control.

L116: no information for secondary antibody for immunostain, callar conjugation.

L148:Figure1 must change to constructed vector pcDNA3.1(+)-Sox18.

Figure6-C. Target should change to β-catenin.

Response 3: Thanks for your comments and we have revised as requested.

1. We show the transfection efficiency under experimental conditions in the results section.

2. We quantified the number of ALP colonies by counting the stained area.

3. We added information about secondary antibodies for immunostain, callar conjugation.

4. We changed Figure 1 to constructed vector pcDNA3.1(+)-Sox18, and placed it in the supplementary material (Figure S1).

5. We changed Target to β-catenin in Figure 6C.

Reviewer 3 Report

The authors studied the function of SOX18 in the growth and development of hair follicles.  There are too many major shortcomings that prevent me from recommending the publication of the current draft. The manuscript was poorly written and should definitely be rewritten before submitting it to any journal!

Major comments:

1) The English in this manuscript looks quite awkward, English editing is required.

2) In biological sciences, “proved” is not the right word to use. “Suggested”, and “indicated” should be used instead.

3) Please do use complete sentences in the. 2. Materials and Methods section.

4) In supplementary materials, detailed results of RNA-seq, etc. should be provided, but the authors only provided Figures from the main text instead.  However, no detailed result of single-cell RNA-seq was provided at all!

5) Figure 1 should be provided as a supplementary figure instead.

6) Please explain why the agglutination of DPCs enhances the induction ability of DPCs?  And please define "induction ability".

7) The authors stated: "Alkaline phosphatase activity is an important index to evaluate the induction ability of DPCs.", please cite a reference o explain why alkaline phosphatase activity can be an important index.

8) What are "third-generation DPCs", how the generation was determined?

Author Response

Response to Reviewer 3:

Point 1: 1) The English in this manuscript looks quite awkward, English editing is required.

Response 1: Thanks for your comments and we thank for your work. We invited English editor to revise the article.

Point 2: 2) In biological sciences, “proved” is not the right word to use. “Suggested”, and “indicated” should be used instead.

Response 2: Thanks for your comments and we have revised as requested.

Point 3: 3) Please do use complete sentences in the. 2. Materials and Methods section.

Response 3: Thanks for your comments and we have revised as requested.

Point 4: 4) In supplementary materials, detailed results of RNA-seq, etc. should be provided, but the authors only provided Figures from the main text instead.  However, no detailed result of single-cell RNA-seq was provided at all!

Response 4: Thanks for your comments. Detailed results of the RNA-seq, etc. have been published in the article “Transcriptomics Reveals the Molecular Anatomy of Sheep Hair Follicle Heterogeneity and Wool Curvature.”, and the relevant data has been uploaded to NCBI’s Gene Expression Omnibus database (https://www.ncbi.nlm.nih.gov/geo/) under ac-cession number: GSE186204. We have added reference in the text.

Point 5: 5) Figure 1 should be provided as a supplementary figure instead.

Response 5: Thanks for your comments and we have revised as requested.

Point 6: 6) Please explain why the agglutination of DPCs enhances the induction ability of DPCs?  And please define "induction ability".

Response 6: Thanks for your comments. The agglutination of DPCs is thought to be related to the induction ability of DPCs and is an indicator to evaluate the induction ability. DPCs with agglutination ability can induce the formation of new hair follicles, but the agglutination of DPCs will gradually disappear. Sox18 enhanced the agglutination of DPCs, so we concluded that Sox18 enhanced the induction ability of DPCs. The induction ability of DPCs means that DPCs are able to induce the formation of hair follicle. We have explained in the text. In future studies, we will demonstrate in vivo of Hu sheep that Sox18 can enhance the induction ability of DPCs.

Point 7: 7) The authors stated: "Alkaline phosphatase activity is an important index to evaluate the induction ability of DPCs.", please cite a reference or explain why alkaline phosphatase activity can be an important index.

Response 7: Thanks for your comments. We cited a reference and made detailed descriptions in the discussion section.

Point 8: 8) What are "third-generation DPCs", how the generation was determined?

Response 8: Thanks for your comments. We used a 0.25% trypsin-EDTA solution to digest the DPCs and considered each digestion of DPCs as one cell passing through to the next generation. We consider primary DPCs that have undergone three trypsin digestions to be third-generation DPCs.